# Quality of life among pregnant women with urinary incontinence: A cross-sectional study in a Malaysian primary care clinic

Aida Jaffar[1,2], Sherina Mohd-Sidik[1]*, Rosliza Abd Manaf[3], Chai Nien Foo[4], Quan Fu Gan[5], Hamimah Saad[6]

**1** Department of Psychiatry, Faculty of Medicine and Health Sciences, Universiti Putra Malaysia, Serdang, Selangor, Malaysia, **2** Primary Care Unit, Faculty of Medicine and Defence Health, Universiti Pertahanan Nasional Malaysia, Sg Besi, Wilayah Persekutuan Kuala Lumpur, Malaysia, **3** Department of Community Health, Faculty of Medicine and Health Sciences, Universiti Putra Malaysia, Serdang, Selangor, Malaysia, **4** Department of Population Medicine, Universiti Tunku Abdul Rahman, Cheras, Kajang, Selangor, Malaysia, **5** Pre-clinical Department, Universiti Tunku Abdul Rahman, Cheras, Kajang, Selangor, Malaysia, **6** Klinik Kesihatan Kajang, Jalan Semenyih, Kajang, Selangor, Malaysia

* sherina@upm.edu.my

## Abstract

### Background

Pregnant women have an increased risk of urinary incontinence (UI), affecting their quality of life (QoL). This study aims to determine UI and its relationship with QoL among incontinent pregnant women.

### Methods

This was a cross-sectional study in a semi-urban primary care clinic in Selangor, Malaysia, among pregnant women aged 18 years old and above. The validated study instruments consisted of questions on socio-demography, the International Consultation on Incontinence Questionnaire-UI Short Form (ICIQ-UI SF) to determine UI and the International Consultation on Incontinence Questionnaire Lower Urinary Tract Symptoms Quality of Life Module (ICIQ-LUTSQoL) to assess their QoL. A generalised linear model was used to determine the association between the continent and incontinent pregnant women with QoL.

### Results

Of the approached 610 respondents, 440 consented to participate in the study, resulting in a response rate of 72.1%. The mean age was 29.8 years old (SD 4.69) with 82.2% (n = 148) having stress UI. Significant independent factors related to the decreased QoL were mid to late trimester (OR 3.06, 95% CI 1.48–6.32), stress UI, (OR 6.94, 95%CI 4.00–12.04) and urge UI (OR3.87, 95%CI 0.48–31.28). Non-Malay improved QoL (OR 0.29, 95% CI 0.16–0.52).

**Data Availability Statement:** All relevant data are within the manuscript and its Supporting information files.

**Funding:** This research was funded by the Universiti Putra Malaysia (UPM/800—3/3/1/GPB/2018/9668500). The funders had no role in study design, data collection and analysis, decision to publish, or preparation of the manuscript.

## Conclusions

All types of UI significantly affecting pregnant women's QoL. This information is useful in enhancing antenatal management at the primary care level, whereby they should be screened for UI and provided with effective early intervention to improve their QoL.

## Introduction

Pregnancy is a challenging process due to multiple changes, for example, a growing uterus that adds additional pressure to the adjacent urinary bladder and pelvic floor muscle and hormonal changes leading to urinary incontinence [1]. Urinary incontinence (UI) is defined by any occasionally urine leakage which can be divided into "Stress" UI (SUI), "Urge" UI (UUI), "Mixed" UI (MUI), Nocturnal enuresis and Post-micturition dribble [2]. SUI is UI when pregnant women coughing or sneezing, UUI is UI with an urgency to void and MUI is the combination of both [2, 3]. Nocturnal enuresis is the UI during sleep and other symptomatic UI are called as post-micturition dribble and continuous urinary leakage [2]. Reviews demonstrated that SUI is the most common UI during pregnancy [3].

A meta-analysis reported that more than half (63%) of incontinent pregnant women with SUI and the weighted average of UI prevalence among pregnant women of 41.0% (CI 95% 34.0–48.0%; I2: 99.77%) [3]. Previously, multiparity has a double increased risk of having UI [4], but recent analysis showed different results. It has reported that nulliparity has slightly higher (42% based on 12 studies) with UI than multiparous (31% based on 4 studies) [3].

More pregnant women experiencing UI with the advancement of the pregnancy as described with 9% in the first trimester (95% CI 6.0–12.0%), 19% in the second trimester (95% CI 12.0–25.0%) and 34% in third trimester (95% CI 23.0–46.0%) [3]. Furthermore, SUI occurred in more than two-thirds in the early (70.4%) and late pregnancy (73.9%) [3]. Therefore, SUI significantly is a risk among pregnant women, and awareness is needed for both the healthcare providers and the pregnant women themselves.

Being overweight and obese during pregnancy added 1.5 times increased risk with 95% Confidence Interval (CI) of 1.28 to 1.83 to have UI compared to normal weight pregnant women [4]. These risks need to be identified during consultations by the healthcare providers, and preventive measures can be taken to reduce the identified risk factors.

Experiencing UI causes pregnant women to go to the toilet frequently even during nighttime which may lead to sleep difficulties and stress that affect their quality of life (QoL). In addition to this, several factors reported to be associated with poorer QoL are obesity, nausea and vomiting, epigastric pain, back pain, and psychological distress [5].

Furthermore, UI may affect their financial expenses as these women may need to buy sanitary absorbent pads for hygienic purposes [6]. A cost analysis study reported that the cost of pads, diapers, laundry, and dry cleaning was about a mean of USD3.91±11.11 per week, leading to an annual cost of USD204±578 in 2007 [7]. Therefore, pregnant women experiencing varieties of loss in their QoL reported with minimum impact [8] to significant impact [9] and socio-economic cost [6, 7].

This study aimed to determine (1) the UI severity among pregnant women, (2) the QoL among pregnant women, and (3) the associations between UI and QoL among pregnant women attended primary care clinic. This study is reporting the third research objective from the previous study [10], and for the need assessment of our future mHealth app interventional study designed to improve UI among pregnant women [11].

## Materials and methods

### Design and respondent selection

A cross-sectional study was conducted from September 2019 till December 2019, at one primary care clinic in one of the nine districts of Selangor, the highest populated state in Malaysia. Informed consent was obtained from the patients and fulfilled the principles established by the Declaration of Helsinki. The approval for this study was obtained from the Medical Research and Ethics Committee (MREC), Ministry of Health Malaysia, and has been registered with the National Medical Research Register (NMRR-19-412-47116) before its implementation. The Ethics Committee for Research Involving Human Subjects, Universiti Putra Malaysia has approved this study with a serial number of JKEUPM-2019-297.

The inclusion criteria were pregnant mothers with a singleton pregnancy, at any trimester, any number of pregnancies (to be as pragmatic approach), able to communicate and read in Malay (the national language of Malaysia) or English. The exclusion criteria for this study were mothers, those with a history of childhood enuresis or pelvic surgery or pelvic organ prolapse, nocturnal enuresis, continuous urinary leakage, history of mental illness or psychosis and history of poorly controlled Diabetes Mellitus.

The sample size calculation was derived from the UI prevalence among pregnant women at 65.8% [12], α = 0.05 and 95% power of the study, giving an estimated sample size of 350 from the S.K. Lwanga and Lemeshow (1991) formula [13]. The final sample size was 440, after considering 20% of non-response.

Simple random sampling method was applied by the enumerators assigned to select patients from the obtained registered patients list at the clinic counter. Therefore, to minimise the researcher bias, briefing sessions and supervision from the researcher and team members were delivered. The respondents signed the consent forms after the study objectives were informed and explained to them, on voluntary purposes. Any respondent could voluntarily withdraw her consent to participate in this study.

### Study instruments

The data instrument for this study was by self-administered questionnaire; for example, socio-demography data, details of the current pregnancy and previous obstetric history.

The self-administered sociodemographic data for example age, ethnicity, occupation, monthly household income, level of education, Body Mass Index (BMI), number of pregnancies, number of the child alive and number of delivery (1) normal delivery, (2) vacuum, (3) forceps (4) Lower Segment Caesarean Section. The BMI was available as the respondents refer to their own antenatal records themselves. Age has been dichotomised with 35 years old based on this study [4].

The International Consultation on Incontinence Questionnaire-Urinary Incontinence Short Form (ICIQ-UI SF) assesses UI frequency, amount of leakage, the overall impact of UI and the type of incontinence and has been validated with the Malay version [14]. ICIQ-UI SF has a Grade A recommendation to diagnose UI and to assess the severity of UI with the impact of UI on pregnant women [2]. This study adapted the severity scoring of slight (1–5), moderate (6–12), severe (13–18), and very severe (19–21) [15].

The International Consultation on Incontinence Questionnaire Urinary Incontinence Quality of Life Module (ICIQ-LUTSQoL) is a recommended QoL questionnaire which was adapted from the King's Health Questionnaire (KHQ) within the ICIQ structure [16]. It delivers a comprehensive measure in assessing the social impact of UI on QoL. It has twenty questions with seven domains and four points answers "1-not at all", "2-slightly", "3-moderate"

and "4-a lot". The seven domains are role limitations, physical limitations, social limitations, personal relationship, emotions, sleep, and severity measures according to each domain's specific items. The overall score is from 19 (not at all) and 76 (a lot), and greater values suggest worsening on QoL [14, 15].

A small pilot study was done among thirty pregnant women before this study to assess the validity and reliability of the Malay version questionnaire. The Cronbach alpha coefficient was 0.622 and 0.916 for the ICIQ-UI SF and ICIQ-LUTSqol, indicating a high level of reliability of this questionnaire in determining the quality of life.

Normality tests were performed before analysing the data for descriptive and inferential statistics. Frequency and percentages were obtained from the frequency statistics and descriptive statistics of numerical data which provided the value of minimum, maximum, mean, and standard deviation, median and interquartile range of the study variables. For inferential analysis, the chi-square test was performed to determine any significant association between the categorical variables.

Bivariate analysis using chi-square test was performed to identify significant associations between the independent variables with the UI severity and QoL of respondents. The predictors of QoL were determined using multiple logistic regression with all variables with $p<0.25$ from the simple logistic regression.

Generalised linear regression was performed to identify significant factors associated with the overall QoL score taking in the variables with a significance level less than 0.25 by bivariate analysis [17]. For all analysis, the significance level was set at $\alpha = 0.05$, and a 95% confidence interval (CI) was applied in this study. The data were analysed using the Statistical Package for Social Science (SPSS) version 25.0 [18].

## Results

Four hundred and forty respondents consented to participate in this study, out of 610 respondents approached. The response rate for this study was 72.1%. One hundred forty pregnant women refused to join the study, and 30 pregnant women repeated respondents. All were included into the analysis as there was no missing data.

The mean age was 29.8 years old (SD 4.69), and the majority were from the Malay ethnicity of 80.9% (n = 356). Two-fifth (40.9%, n = 180) of them reported with UI, 52.8% (n = 95) reported slight UI, 44.4% (n = 80) moderate UI and 2.8% (n = 5) severe UI which the detail has been reported in our recent publication [10]. For the UI severity, we grouped them into slight UI, and moderate UI, whereby we combined moderate and severe UI into one group as the number of severe UI was too small to be analysed independently.

The total of 170 respondents was at their first-time pregnancy, and two-third of respondent had vaginal delivery (71.1%, n = 192) and about a third of them (28.9%, n = 78) had previous lower segment caesarean section (LSCS) before.

Slight and moderate UI significantly associated with pregnant women less than 35 years age (p = 0.05) and at their $2^{nd}$–$5^{th}$ gravida (p = 0.001). Pregnant women at their third trimester (p = 0.04) and history of vaginal delivery (p = 0.05) were significantly associated with slight and moderate UI as shown in Table 1.

The QoL was categorized according to the median score which was 23 in this study [9]. The score less than 23 was categorized as good QoL and 23 and above as poor QoL. Table 2 listed the significant associations with their QoL. Poor QoL among respondents was significantly associated with Malay ethnicity (p = <0.001), advancement trimester (p = 0.002) and all the types of UI with stress UI, mixed UI and urge UI (p<0.001).

**Table 1. Demographics of the study respondents (N = 440).**

| Variables | UI Severity | | | $X^2$ | P |
|---|---|---|---|---|---|
| | No UI (N = 260) n (%) | Slight UI (N = 95) n (%) | Mod UI (N = 85) n (%) | | |
| **Age** | | | | 6.00 | 0.05* |
| Less than 35 | 225 (61.3) | 78 (21.3) | 64 (17.4) | | |
| 35 and above | 35 (47.9) | 17 (23.3) | 21 (28.8) | | |
| **Education** | | | | 3.744[#] | 0.581[#] |
| Primary | 3 (75.0) | 0 (0) | 1 (25.0) | | |
| Secondary | 86 (62.8) | 29 (21.3) | 22 (15.9) | | |
| Tertiary | 168 (56.9) | 65 (22.0) | 62 (21.1) | | |
| **Ethnicity** | | | | 5.01 | 0.08 |
| Malay | 202 (56.7) | 79 (22.2) | 75 (21.2) | | |
| Non-Malay | 58 (69.0) | 16 (19.0) | 10 (12.0) | | |
| **Income** | | | | 1.74 | 0.42 |
| <RM3000 | 179 (58.9) | 69 (22.7) | 56 (18.4) | | |
| ≥RM3000 | 37 (55.2) | 13 (19.4) | 17 (25.4) | | |
| **BMI** | | | | 7.47 | 0.11 |
| Normal and below | 123 (66.1) | 35 (18.8) | 28 (15.1) | | |
| Overweight | 80 (55.6) | 34 (23.6) | 30 (20.8) | | |
| Obese | 57 (51.8) | 26 (23.6) | 27 (24.6) | | |
| **Gravida** | | | | 17.58 | 0.001* |
| 1 | 120 (70.6) | 31 (18.2) | 19 (11.2) | | |
| 2–5 | 120 (52.2) | 53 (23.0) | 57 (24.8) | | |
| >5 | 20 (50.0) | 11 (27.5) | 9 (22.5) | | |
| **Trimester** | | | | 9.59 | 0.04* |
| First | 38 (71.7) | 8 (15.1) | 7 (13.2) | | |
| Second | 87 (57.2) | 27 (17.8) | 38 (25.0) | | |
| Third | 135 (57.4) | 60 (25.5) | 40 (17.1) | | |
| **Obstetric History**[$] | | | | 5.9 | 0.05* |
| Vaginal delivery | 101 (52.6) | 51 (26.6) | 40 (20.8) | | |
| LSCS | 39 (50.0) | 13 (16.7) | 26 (33.3) | | |

Moderate severity consists of moderate and severe UI.

LSCS-Lower Segment Caesarean Section.

[$]Obstetric history from a total of 270 respondents multigravida.

[#]Fisher exact test.

*Statistically significant.

Based on a p-value of 0.25 from simple logistic regression, age, ethnicity, BMI, trimester, stress UI, mixed UI and urge UI were entered into the multiple logistic regression analysis model to predict the poor QoL among respondents (Table 3). The omnibus model for the logistic regression analysis was statistically significant, $X^2$ ($df$ = 7, N = 339) = 133.058, $p<0.001$, Cox and Snell $R^2$ = 0.261, Nagelkerke $R^2$ = 0.349. The model was 71.8% accurate in its prediction of poor QoL. The Hosmer and Lemeshow test confirmed that the model was good fit for the data.

Pregnant women with mid to late trimester have three times odd with decreased QoL (OR 3.06, 95% CI 1.48–6.32). However, being Non-Malay, has improved QoL (OR 0.29, 95% CI 0.16–0.52). Unfortunately, for pregnant women who have stress UI, they have six times

**Table 2. Associations with quality of life of the respondents (N = 440).**

| Variables | Quality of Life | | $X^2$ | P |
|---|---|---|---|---|
| | Good n(%) | Poor n(%) | | |
| **Age** | | | 2.132 | 0.09 |
| Less than 35 | 175 (47.7) | 192 (52.3) | | |
| 35 and above | 28 (38.4) | 45 (61.6) | | |
| **Education** | | | 1.433 | 0.488 |
| Primary | 3 (75.0) | 1(25.0) | | |
| Secondary | 63 (46.0) | 74 (54.0) | | |
| Tertiary | 133 (45.1) | 162 (54.9) | | |
| **Ethnicity** | | | 24.268 | <0.001 (Phi -0.235) |
| Malay | 144 (40.4) | 212 (59.6) | | |
| Non-Malay | 59 (70.2) | 25 (29.8) | | |
| **Income** | | | 0.388 | 0.533 |
| <RM3000 | 137 (45.1) | 167 (54.9) | | |
| ≥RM3000 | 33 (49.3) | 34 (50.7) | | |
| **BMI** | | | 1.935 | 0.176 |
| Normal and below | 93 (50.0) | 93 (50.0) | | |
| Overweight and obese | 110 (43.3) | 144 (56.7) | | |
| **Gravida** | | | 0.491 | 0.493 |
| 1 | 82 (48.2) | 88 (51.8) | | |
| >2 | 121 (44.8) | 149 (55.2) | | |
| **Trimester** | | | 9.604 | 0.002 (Phi 0.148) |
| First | 35 (66.0) | 18 (34.0) | | |
| Second-Third | 168 (43.4) | 219 (56.6) | | |
| **Obstetric History** [$] | | | 1.006 | 0.316 |
| Vaginal delivery | 74 (38.5) | 118 (61.5) | | |
| LSCS | 25 (32.1) | 53 (67.9) | | |
| **Stress UI** | | | 61.014 | <0.001 (Phi 0.372) |
| No | 178 (58.6) | 126 (41.4) | | |
| Yes | 25 (18.4) | 111 (81.6) | | |
| **Mixed UI** | | | 12.368 | <0.001 (Phi 0.168) |
| No | 201 (47.7) | 220 (52.3) | | |
| Yes | 1 (5.6) | 17 (94.4) | | |
| **Urge UI** | | | 22.837 | <0.001 (Phi 0.228) |
| No | 198 (49.9) | 199 (50.1) | | |
| Yes | 5 (11.6) | 38 (88.4) | | |

[$]Obstetric history from a total of 270 respondents multigravida.

increased odds (OR 6.94, 95%CI 4.00–12.04) and those with urge UI have three times increased odds (OR3.87, 95%CI 0.48–31.28) to have a decreased QoL (Table 3).

The same variables were further analysed to determine their associations with the domain of QoL in Table 4. Using the same ICIQ-LUTSQoL seven domains; (1) role limitation, (2) physical limitation, (3) social limitation, (4) personal relationship, (5) emotion limitation, (6) sleep/emotion limitation, and (7) severity measures were included in the Generalised Linear Model regression.

**Table 3. Multivariate analysis of the significant predictors of QoL among pregnant women.**

| Predictors | Multivariate analysis | | | |
|---|---|---|---|---|
| | β[a] | SE[b] | Exp(β) [95% CI[c]] | p |
| Age > 35 years | 0.09 | 0.307 | 1.094 [0.600–1.995] | 0.769 |
| Non-Malay | -1.249 | 0.302 | 0.287 [0.159–0.518] | <0.001 |
| Mid-Late Trimester | 1.119 | 0.369 | 3.062 [1.484–6.317] | 0.002 |
| Stress UI | 1.937 | .281 | 6.941 [4.002–12.037] | <0.001 |
| Mixed UI | 1.353 | 1.066 | 3.868 [0.478–31.277] | 0.205 |
| Urge UI | 2.433 | 0.507 | 11.390[4.215–30.774] | <0.001 |

Sleep limitation were significantly associated among mid-late trimester respondents (p = 0.044) and respondents from the high educational status (p = 0.026). In contrast, being non-Malay, they were not associated with role limitation (p = 0.007), physical limitation (p = 0.012), social limitation (p = 0.019) and severity measures (p<0.001).

Among the three types of UI, mixed UI did not show any significant relationship with limitation to their personal relationship and emotion. Unfortunately, respondents with stress UI and urge UI has significant association with all domains of decreased QoL.

## Discussion

Malay pregnant women, women in their second or third trimester, and women with stress UI or urge UI, had greater odds of poor incontinence specific quality of life. Variables not associated with worse incontinence-specific quality of life were age >35 years old, and mixed UI. The most affected QoL domains were severity measures.

Pregnant women aged 35 years and above demonstrated a significant association in UI in this study. A meta-analysis reported a 1.5 added risk of UI (95% CI:1.45 to 1.62) among pregnant women once they are 35 years with a cautious note from the authors as the studies were

**Table 4. Multivariate analysis of the significant predictors of QoL among pregnant women.**

| Variables | Beta coefficient (95% Confidence Interval) | | | | | | |
|---|---|---|---|---|---|---|---|
| | Role Limitation | Physical Limitation | Social Limitation | Personal Relationship | Emotion Limitation | Sleep/ Energy Limitation | Severity Measure |
| **Age**: 35 and above | 5.538 (0.897, 10.180) | 4.116 (-1.000,9.234) | 2.248 (-1.505,6.001) | 2.748 (-1.566,7.062) | -0.969 (-4.188,2.250) | 0.258 (-4.278,4.793) | -2.170 (-7.112,2.773) |
| P value | 0.019* | 0.115 | 0.240 | 0.212 | 0.555 | 0.911 | 0.390 |
| **Ethnicity**: Non-Malay | -5.985 (-10.30,-1.669) | -6.027 (-10.731,1.324) | -4.143 (-7.593,-0.693) | -2.718 (-6.683,1.248) | -.475 (-3.433, 2.484) | -3.344 (-7.513,0.825) | -8.783 (-13.326,4.240) |
| P value | 0.007* | 0.012* | 0.019* | 0.179 | 0.753 | 0.116 | <0.001* |
| **Trimester**: Mid-Late | 4.456 (-.705,9.617) | 2.779 (-2.778,8.337) | -0.117 (-4.193,3.958) | 0.812 (-3.873,5.497) | -0.044 (-3.539,3.452) | 5.061 (0.135,9.987) | 7.645 (2.278,13.012) |
| P Value | 0.091 | 0.327 | 0.955 | 0.734 | 0.980 | 0.044* | 0.005* |
| **Stress UI** | 20.693 (16.849,24.537) | 17.586 (13.355,21.817) | 9.366 (6.263,12.469) | 7.323 (3.756,10.891) | 8.506 (5.845,11.168) | 7.344 (3.594,11.095) | 9.815 (5.728,13.902) |
| P Value | <0.001* | <0.001* | <0.001* | <0.001* | <0.001* | <0.001* | <0.001* |
| **Mixed UI** | 12.704 (3.771, 21.637) | 17.807 (8.181,27.432) | 7.087 (0.027,14.146) | 5.822 (-2.292,13.937) | 4.169 (-1.885,10.223) | 8.988 (0.457,17.520) | 17.559 (8.262,26.855) |
| P Value | 0.005* | <0.001* | 0.049* | 0.16 | 0.177 | 0.039* | <0.001* |
| **Urge UI** | 18.131 (12.540,23.723) | 17.916 (11.892,23.940) | 8.201 (3.783,12.619) | 6.898 (1.820, 11.977) | 9.682 (5.893,13.471) | 11.647 (6.307,16.987) | 15.525 (9.706,21.343) |
| P Value | <0.001* | <0.001* | <0.001* | 0.008* | <0.001* | <0.001* | <0.001* |

low-quality studies [4]. Another local study at tertiary care centre reported that primigravida more than 30-year-old with 48.1% (n = 13/27) has UI [19]. However, the same study did not report any significant association between age and UI among primigravida [19]. Abdullah et. al., reported with two significant association UI among the primigravida: childhood enuresis (p = 0.003), and previous history of urinary incontinence (p < 0.001) [19]. Unfortunately, this study was unable to compare these association as we did not examine these factors.

Multigravida in this study was significantly associated with moderate UI (p = 0.001); this signifies the importance of parity in predicting UI, similar to a meta-analysis report [4]. Pelvic floor muscle strength of multigravida women was reduced at 22–35% starting at 20 weeks pregnancy until six weeks postpartum [1]. The reason for this could be due to a reduction in collagen which plays an essential role in the tensile properties of the pelvic floor muscle [1]. Reduction of collagen further decreases the joint laxity and stretching of pelvic ligaments capability leading to an impairment of the pelvic floor muscle support causing pregnant women to experience UI. Interestingly, previous local studies did not find any significant association with UI among both primigravida and in combinations at tertiary hospital respondents [12, 19].

UI worsened with trimester advancement (p = 0.04) due to the direct pressure on the bladder from the increasing fetal weight and expansion of the uterus [1]. In contrast, previous local studies among primigravida [19] and mixed gravida [12] did not show any significant association between UI with the trimester. This could be both studies were done at the tertiary centre and this study was done at primary care centre. Their study respondents were from the primary care referrals and more complicated when compared to this study. This may signify the homogeneous complexity of their study background when compared to this study where the complicated cases have been excluded.

Vaginal deliveries were significantly associated with moderate UI (p = 0.002) in this study. Childbirth process can damage the pubovisceral muscle and surrounding fascia with nerve disruption causing reduction of the pelvic floor muscle strength [20, 21]. This finding correlates with findings from a Norway primigravida prospective study among 976 pregnant women which reported that two-thirds of their respondents (77.4%, n = 222/304) had UI at 12 months postpartum after spontaneous vaginal deliveries and only 6.7% (n = 20/300) had UI post-caesarean section deliveries [22].

Therefore, this study reported with more significant associations when compare with the previous local studies. Abdullah et. al, studied UI focusing among primigravida and Yusof et. al, studied UI at any times of pregnancy but using different UI assessment questionnaire "Revised Urinary Incontinence Scale (RUIS)" [12, 19]. This suggests that pregnant women at primary care clinic level may have higher risk of UI when compared to the pregnant women at tertiary care centre with other medical diagnoses.

The QoL domains such as "role limitation", "physical limitation" "personal relationship emotion" and less problems with "personal relationship emotions", "sleep energy" and "severity measures" were assessed. This study found that all domains were significantly affected by pregnant women with all types of UI, specifically in the severity measures (wearing pad, cautious water intake, changing wet underclothes and worries about own smell), physical, and role limitations.

Unlike with the mixed UI group which limiting their role, physical, social, sleep/energy and severity measures, pregnant women with Stress UI and Urge UI experiencing global limitation. Previous study reported that UI affecting pregnant women mild to moderately [3] which similar to the respondents in this study.

Pregnant women try to limit their fluid intake as more intakes will lead to a risk of UI. They avoid caffeinated drinks to ensure less frequency to the toilet. Also, they are worried about

travelling to work as they need to hold their bladder for quite some time. They wear pads to ensure they are dry, but at the same time, they were worried about the urine smell, especially at the workplace or in transportation. Some of them feel embarrassment due to UI and have low self-esteem [23].

A cross-sectional study with 261 pregnant women in Brazil reported three predictors which negatively influenced their QoL; occupation/work (p < .001), polyuria (p = .004) and fatigue (p = .006) [24]. Pregnant women with UI, had to go to the toilet more frequently, especially at night, which disturbed their sleep pattern. Since adequate sleep is crucial for women at all pregnancy stages for an optimal health related QoL, particularly later in pregnancy, sleep disturbance can lead to more stress and postpartum depression. Similarly, with our study, sleep was also affected especially among educated pregnant women, mid-late trimester, and incontinent pregnant women. They needed to go immediately to the toilet as they had a high risk of incontinence when they have the urge to urinate.

## Implication for practice

This study informs both the healthcare providers and pregnant women especially Malay ethnicity has poor QoL. Pregnant women with the advancement of trimester and from the educated background has sleep limitation. All pregnant women should be aware that they will experiencing sleep disturbance and to prepare measures to improve their sleep quality. Educated pregnant women who drives to work should be aware on their limitation and to ensure to have good quality of sleep to avoid road traffic accident when driving to work.

Pregnant women with UI have experience limitation to all domain in QoL. The social limitation will add their stress level and risk of psychological distress. Their role limitation which may affect their ability to do their daily task and personal relationship limitation which may affecting their spouse. They need to spend more for the absorbent pad, and they worried of urine smell.

Therefore, healthcare providers must consider educating all pregnant women regarding importance of QoL and risk of UI during their pregnancy. UI as an essential condition to be dealt with clinically, instead of accepting it as a norm and addressing it by traditional means. Healthcare providers should screen the pregnant women even at their first trimester and among primigravida. All pregnant women should be aware on the UI and the conservative management available which is pelvic floor muscle training to enhance pregnant women's quality of life.

## Study limitation

Data collection using self-report is major limitation in this study as we could not measure objectively the UI. Furthermore, conducting in a single-centre primary care which affecting its external validity. However, the study findings will be acceptable for our need assessment of future interventional study [11].

## Conclusions

The findings of this study demonstrate that pregnant women, at primary care level, even during the first trimester and primigravida, have a risk of moderately severe UI throughout pregnancy. Pregnant women with mid-late trimester and educated background has poor sleep. Having UI worsen their QoL, especially for pregnant woman who is having stress UI or urge UI. These findings are useful to increase awareness among pregnant women and health care providers.

## Supporting information

**S1 File.**
(PDF)

**S1 Data.**
(XLSX)

## Acknowledgments

Permission to publish this paper was granted by the Medical Research and Ethics Committee (MREC), Ministry of Health Malaysia. The authors would like to thank the Director General of Health, Ministry of Health, Malaysia for permission to publish this paper. We would like to acknowledge the students (Chan Khai Ying, Ho Jing Jie, Lim Ming Yi, Lim Yee Xien, Teoh Yong Xing and Nurul 'Afifah Md Ali), the clinic's staff nurses, sisters and matron involved in the data collection. The authors would like to thank the respondents involved in the study.

## Author Contributions

**Conceptualization:** Aida Jaffar, Hamimah Saad.

**Data curation:** Aida Jaffar, Rosliza Abd Manaf, Chai Nien Foo.

**Formal analysis:** Aida Jaffar.

**Funding acquisition:** Sherina Mohd-Sidik.

**Investigation:** Chai Nien Foo, Quan Fu Gan.

**Methodology:** Aida Jaffar.

**Project administration:** Aida Jaffar, Sherina Mohd-Sidik, Chai Nien Foo.

**Resources:** Aida Jaffar, Sherina Mohd-Sidik.

**Supervision:** Sherina Mohd-Sidik.

**Visualization:** Aida Jaffar.

**Writing – original draft:** Aida Jaffar, Chai Nien Foo, Quan Fu Gan.

**Writing – review & editing:** Sherina Mohd-Sidik, Rosliza Abd Manaf, Hamimah Saad.

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
