## [Decision Letter · Decision Letter 0]

11 Feb 2021

PONE-D-20-35389

What is the quality of life among pregnant women with urinary incontinence?

PLOS ONE

Dear Dr. Mohd-Sidik,

Thank you for submitting your manuscript to PLOS ONE. After careful consideration, we feel that it has merit but does not fully meet PLOS ONE’s publication criteria as it currently stands. Therefore, we invite you to submit a revised version of the manuscript that addresses the points raised during the review process.

We look forward to receiving your revised manuscript.

Kind regards,

Peter F.W.M. Rosier, M.D. PhD

Academic Editor

PLOS ONE

Additional Editor Comments:

I think you have received very useful comments from the reviewers. I advise you to make good use of it.

Journal Requirements:

2.) Please include additional information regarding the survey or questionnaire used in the study and ensure that you have provided sufficient details that others could replicate the analyses. For instance, if you developed a questionnaire as part of this study and it is not under a copyright more restrictive than CC-BY, please include a copy, in both the original language and English, as Supporting Information.

3.) Please ensure you have thoroughly discussed any potential limitations of this study within the Discussion section, including the potential impact of confounding factors.

4.) Please include captions for your Supporting Information files at the end of your manuscript, and update any in-text citations to match accordingly. Please see our Supporting Information guidelines for more information: http://journals.plos.org/plosone/s/supporting-information.

Reviewers' comments:

Reviewer's Responses to Questions

**Comments to the Author**

1. Is the manuscript technically sound, and do the data support the conclusions?

Reviewer #1: Yes

Reviewer #2: Yes

Reviewer #3: Partly

2. Has the statistical analysis been performed appropriately and rigorously? 

Reviewer #1: I Don't Know

Reviewer #2: Yes

Reviewer #3: Yes

3. Have the authors made all data underlying the findings in their manuscript fully available?

Reviewer #1: No

Reviewer #2: Yes

Reviewer #3: Yes

4. Is the manuscript presented in an intelligible fashion and written in standard English?

Reviewer #1: No

Reviewer #2: Yes

Reviewer #3: Yes

5. Review Comments to the Author

Reviewer #1: This is an appropriate study to look at the quality of life in pregnant women with urinary incontinence in a local Malaysian community

Major

i) Introduction: The author to consider reviewing the definitions of the different types of UI and to address all different types of urinary incontinence. Then to consider in the methods section under in and exclusion criteria which types were excluded.

ii)Methods: Author to clarify if the QOL questionnaire (ICIQ-LUTSQoL) was validated for their population and if so in which languages. The validation and local validation of their measure instrument (questionnaires) should reflect in the abstract and text methods section.

iii) Results:

Question to author is why author prefers to define and use the term urine “status” and not ”severity” of urinary incontinence? The severity is then categorized in no UI, slight and moderate incontinence where moderate includes severe urinary incontinence. If the author, consider changing the terminology it may contribute to easier reading.

Page 8: Line 14: Consider replacing “status” with severity as well as in the Table1)

Line 14: The presentation of the association is not clear to the reader. Can this be rephrased for more clarity to the reader?

a) Page 9 Table 1: Suggest the author consider completing the table1 in total which may inform the reader better. Also indicate in the table Moderate and Severe incontinence is combined

iv) Discussion:

The author did not clearly address the strength and limitations of their study

The author to consider addressing the above.

Page 15 line 6: Is it true that damaged pelvic muscles causes urethral descent and mobilization? Can there be another explanation such as pelvic fascia or ligaments?

Minor

Page 8: line 9: Total of 180 respondents ……. severe UI. Consider reviewing as it makes out difficult reading

Page 9 Line 4-6 (Grammar): The study …… assessing the QOL. Age ….

Page 9 Line 4-6: Author may consider indicating if associations are associated with decreased or improved quality of life?

Page 13 line 13: This study……. with moderate UI …. negative Qol. This sentence needs to be reviewed / rephrased to improve clarity to the reader

Reviewer #2: First of all, thank you very much for give me the best opportunity for taking the reviewer role to review this manuscript. This is an interesting paper. The authors present a study to assess the risk factors and impact of UI on the QoL among incontinent pregnant women. A cross-sectional study design was used.

I have read carefully and found that this study is very carefully created and developed. Although this study has scientific interest, some important aspect should be reviewed by the authors. I hope that my opinions will help shape your research article more precise and interesting. The followings are my comments;

1. The title of the study is not completely clear. The title indicates the dependent variable and samples. But, the independent variable and study design have not been included into the title. This makes the article sounded like a “review article” more than “research article”. Therefore, the authors should add the independent variable and study design in the title and rewrite the title.

2. The authors presented an appropriate and clear detail about the abstract section, but some point should add in topics as following: the authors please add sample size of the participants and duration of data collection.

3. The authors used appropriate key words. However, the authors please add “risk factors” as the key words of the study to guide the reader to easily find a good research title and attract to read it.

4. The introduction of the study is good written and shows the significance of the study. The author wrote the introduction in orderly manner beginning from relationship between pregnancy and urinary incontinence, definition, prevalence, risk factors, affect QoL, limitation of previous studies and aims of the study. Moreover, the authors presented a clear state of the aims of the study, but not showed the congruence with the aims of the study in the abstract section.

5. The authors used a cross-sectional study as a study design. It is an appropriate design in this study. The setting is clear. The authors presented a clear state of sample, inclusion and exclusion criteria, duration of data collection, sample size calculation and sampling method. Moreover, the protection of human subjects was clearly stated and presented the number of IRB (JKEUPM-2019-297) in the section. All subjects gave written informed consent before entering the study.

6. The authors presented a clear state of the measurements of the study which included the details, scoring, and classification of all measurements in the study (ICIQ-UI SF and ICIQ-LUTSQoL). However, what is the reliability of ICIQ-UI SF and ICIQ-LUTSQoL in the study? The authors please state the reliability of all measurements in the study.

7. In the results section, the authors presented all data both text and tables. At the beginning, the authors detailed about the demographic of the pregnant women with and without UI during pregnancy which presented in Table 1. Consequently, the authors used three tables to present the results of the study as following: Table 2 to present the scores of QoL, Table 3 to present seven domains of QoL with significant sociodemographics and Types of UI, and Table 4 to present the significant predictors of QoL among pregnant women. Therefore, the authors used appropriate number and running head title of the tables to report the significant findings, these may help the readers more clearly understand.

8. The discussion had been good written which show the consistency of the aims and the results of the study. Moreover, the authors discussed the results of the study comparing with the previous published studies. It may help the readers more clearly understanding.

9. The authors clearly described the implication for practice of the study.

10. The authors did not state the strengths and limitations of the study. Please state the strengths and limitations of the study.

11. The authors clearly presented conclusion and followed logically from the results of the study

12. The references that the authors cited in the text were published in the high standard journals in urogynecology field and had high relevance to the study which the authors interested in UI in the pregnant women. Moreover, most of references which the authors cited in the study were not over ten years. There were assumed that the knowledge from the previous published articles is not out of date. However, the authors please check the correct format of citations and references based on PLOS ONE.

Reviewer #3: I have used the STROBE statement to guide my review (Vandenbroucke JP, Von Elm E, Altman DG, Gøtzsche PC, Mulrow CD, Pocock SJ, Poole C, Schlesselman JJ, Egger M, Strobe Initiative. Strengthening the Reporting of Observational Studies in Epidemiology (STROBE): explanation and elaboration. PLoS Med. 2007 Oct 16;4(10):e297.)

Item 1a. study design not in title. Make it clear that the study investigates incontinence-specific QoL

Item 1b. revise when comments addressed.

Item 2. Does not make it clear that there is QoL (as an ‘overall’ concept) and incontinence-specific QoL. When you report QoL data in the introduction it is not clear which of these you refer to. Paragraph 5 of the introduction says there are limited reports of UI in pregnant women in Asia. This is insufficient. What is the body of evidence on this topic in a similar population? I note at least one publication in the reference list that is directly relevant. Why is a study needed? What can your study add?

Item 3. Study objective not clearly phrased. Seems to imply only interested in QoL in incontinent women, but the study cohort is women with and without incontinence. This is also the place to make it clear that you are investigating incontinence specific QoL

Items 7 and 8 and 11 and 16b. Please explain every variable (e.g. what are the sociodemographic and obstetric variables) and in what form the data were collected (e.g. date of birth), transformed (e.g. age at questionnaire completion) and categorised for analysis (e.g. dichotomisation into under 35 years of age, and 35 and over). Please also justify all categorisations (e.g. why 35?). Were all data self-reported including data such as gravidity and body weight and height (both of which you must have needed to calculate BMI) or were some data taken from the clinic record? Remember to discuss the accuracy of self-reported data (e.g. for weight/height) as this might influence your findings. The response categories on the ICIQ-LUTSQoL are ordinal yet the analysis seems is completed as if the data are continuous. Please explain/justify.

Item 12c. Were there any missing data? If yes, explain how this was handled. If none, I think it is worth saying so.

Item 13. What do you know about the women who did not take part (e.g. demographics? Reasons for non-participation?)

Item 14. Table 1 presents the participant description by continence status (none, mild, moderate). Thus, there is no reporting of the descriptive data before it was categorised in some way.

Item 16. The presentation of results in the text (the tables are generally good) lacks clarity throughout. For instance, the authors say “There was a clear association between the status of UI according to its severity and age, gravida, trimester, and vaginal delivery”. This tells me nothing about the direction of association, and UI status (e.g. is it younger or older age that is associated with having which UI severity?) And, in table 1 explain how the delivery mode data add up to n=440. If some of the women in this study are in their first pregnancy (n=170?) then they have not had a delivery yet. In tables 2 and 3 the mean difference and 95% CI for the difference is missing – p values are not sufficient. Without a mean difference and confidence interval it is very difficult to interpret this relative to the scoring of the QoL instrument. Why does table 2 not have delivery mode? Why are there a number of variables missing (e.g. age, education, etc) from table 3? This starts to appear like selective reporting. Why is age missing from the regression model? – based on table 2 it looks like it should be included.

Items 18 to 21. Please address the study objective – clearly summarise the findings. For instance, ‘Non-Malay pregnant women, women in their second or third trimester, and women with stress UI, had greater odds of poor incontinence-specific quality of life. Variables not associated with worse incontinence-specific quality of life were…………………The most affected QoL domains were……” There is no discussion of any study limitations – what were they and what influence might they have on your findings? Please put the findings in the context of the data from other study on UI prevalence in pregnant women from Malaysia – what does your study add or confirm, and are there any inconsistencies and how might they be explained? Do the associations you find fit the pattern of associations others have found (or not)? How generalizable are these data? Most of the discussion about needs to inform women about various things irrelevant as your study objective is not about educating women or about interventions for UI. The implications for practice are not relevant to this study either. Only the first paragraph in the implications section could be considered appropriate – e.g. if it is common and affecting QoL then screening might be important – based on your data what can you say about who and how and when to find out if women have UI and what impact this is having for them?

Minor editorial

Introduction. Para 1. Last sentence. Reviews of what?

Introduction. Para 6. Last sentence. Link between the two concepts (QoL, PFMT) not made clear. What do they have to do with each other?

Design and respondent selection. Para 2. Mothers with. Not “mothers, those with”. And para 4. The word enumerators is not commonly used in English. Do you mean research assistants? And also in para 4, do you mean a woman could decline to participate or do you mean she could consent and then having consent withdraw? If women could withdraw once consented what happened to their data?

Study instruments. Para 1. Last sentence - Do you adapted or adopted. If you adapted a measure in any way this must be explained fully and justified.

Discussion. Don’t repeat data in the discussion.

Reference. Multiple errors and inconsistencies. Needs thorough check.

Cannot understand these sentences or phrases:

Introduction. Para 5. First sentence. “the varieties of QOL reported minimum impact”

Design and respondent selection. Para 4. “on voluntary purposes”.

Results. Para 3. “All significant variables significantly associated role limitation and severity measures” Note, significantly is spelt incorrectly too.

Results. Para 4. “Double protective risk of having poor QoL”

6. PLOS authors have the option to publish the peer review history of their article (what does this mean?). If published, this will include your full peer review and any attached files.

Reviewer #1: No

Reviewer #2: No

Reviewer #3: No

---

## [Author Response · Author response to Decision Letter 0]

1 Mar 2021

Reviewer #1

This is an appropriate study to look at the quality of life in pregnant women with urinary incontinence in a local Malaysian community.

Author’s comment: We appreciate the reviewer’s opinion on this study; indeed, this is an appropriate study for our local population. This manuscript will be a reference for future study among pregnant women.

Major

i) Introduction: The author to consider reviewing the definitions of the different types of UI and to address all different types of urinary incontinence. Then to consider in the methods section under in and exclusion criteria which types were excluded.

Authors’ comment: We thank for this feedback. We have revised the definitions and we added recent meta analysis as reference. We include all types of UI according to the ICIQ-UI SF in this study and we have add the nocturnal enuresis and continuous urinary leakage at the exclusion criteria. “SUI is UI when pregnant women coughing or sneezing, UUI is UI with an urgency to void and MUI is the combination of both. Nocturnal enuresis is the UI during sleep and other symptomatic UI are called as Post-micturition dribble and continuous urinary leakage.” (Page 4 Para 1) 

Moossdorff-Steinhauser HFA, Berghmans BCM, Spaanderman MEA, Bols EMJ. Prevalence, incidence and bothersomeness of urinary incontinence in pregnancy: a systematic review and meta-analysis. Int Urogynecol J [Internet]. 2021; Available from: https://doi.org/10.1007/s00192-020-04636-3

ii)Methods: Author to clarify if the QOL questionnaire (ICIQ-LUTSQoL) was validated for their population and if so in which languages. The validation and local validation of their measure instrument (questionnaires) should reflect in the abstract and text methods section.

Authors’ comment: We appreciate the reviewer’s concern with the validation and reliability of the questionnaire. We added as below:

“A small pilot study was done among thirty pregnant women before this study to assess the validity and reliability of the Malay version questionnaire (13). The Cronbach alpha coefficient was 0.622 and 0.916 for the ICIQ-UI SF and ICIQ-LUTSqol, respectively, which indicates a high level of reliability of this questionnaire in determining the quality of life.” (Page 8 Para 2)

iii) Results:

Question to author is why author prefers to define and use the term urine “status” and not ”severity” of urinary incontinence? The severity is then categorized in no UI, slight and moderate incontinence where moderate includes severe urinary incontinence. If the author, consider changing the terminology it may contribute to easier reading.

Authors’ comment: We appreciate this suggestion. We replace the status is the manuscript with the severity. Thank you very much for this insightful suggestion.

Page 8: Line 14: Consider replacing “status” with severity as well as in the Table1)

Authors’ comment: We agree, and we replace all the “status” of the UI in the manuscript with the severity. Thank you very much.

Line 14: The presentation of the association is not clear to the reader. Can this be rephrased for more clarity to the reader?

Authors’ comment: We are thankful for the reviewer’s feedback. We replace with another sentence as follow:

“Slight and moderate UI significantly associated with pregnant women less than 35 years age (p=0.05) and at their 2nd - 5th gravida (p=0.001). Pregnant women at their third trimester (p=0.04) and history of vaginal delivery (p=0.05) were significantly associated with slight and moderate UI as shown in Table 1.” (Page 9, Para 4)

a) Page 9 Table 1: Suggest the author consider completing the table1 in total which may inform the reader better. Also indicate in the table Moderate and Severe incontinence is combined.

Authors’ comment: We appreciate, and we have corrected according to the reviewer’s suggestion.

iv) Discussion:

The author did not clearly address the strength and limitations of their study.

Author’s comment: We would like to thank you for this comment. We admit that we do miss to add the study limitation at the end of the discussion. We added the study limitation as below. 

“This study has been done in a single-centre primary care which affecting its external validity. The cross-sectional design declines the ability to conclude its temporal relationship. However, the study findings will be acceptable for our need assessment of future interventional study (12).” (Page 18 Para 4)

The author to consider addressing the above.

Page 15 line 6: Is it true that damaged pelvic muscles causes urethral descent and mobilization? Can there be another explanation such as pelvic fascia or ligaments?

Author’s comment: We would like to thank you for this insightful suggestion. We have add as “Childbirth process can damage the pubovisceral muscle and surrounding fascia with nerve disruption causing reduction of the pelvic floor muscle strength.” (Page 16 Para 2)

Ashton-Miller J.A., DeLancey J.O.L. (2021) Mechanisms of Pelvic Floor Trauma During Vaginal Delivery. In: Santoro G.A., Wieczorek A.P., Sultan A.H. (eds) Pelvic Floor Disorders. Springer, Cham. https://doi.org/10.1007/978-3-030-40862-6_12

Sultan A.H., Thakar R. (2021) Posterior Compartment Trauma and Management of Acute Obstetric Anal Sphincter Injuries. In: Santoro G.A., Wieczorek A.P., Sultan A.H. (eds) Pelvic Floor Disorders. Springer, Cham. https://doi.org/10.1007/978-3-030-40862-6_13

Minor

Page 8: line 9: Total of 180 respondents ……. severe UI. Consider reviewing as it makes out difficult reading

Authors’ comment: We do agree with the comment. We do feel it is not easy to understand; hence we rephrase as below.

“Two-fifth (40.9%, n=180) of them reported with UI, 52.8% (n=95) reported slight UI, 44.4% (n=80) moderate UI and 2.8% (n=5) severe UI which the detail has been reported in our recent publication (10).” (Page 9 Para 2)

Page 9 Line 4-6 (Grammar): The study …… assessing the QOL. Age ….

Authors’ comment: We do agree with the grammatical issue of the sentence; hence we rephrase as below.

The QoL was categorized according to the median score which was 23 in this study (9). The score less than 23 was categorized as good QoL and 23 and above as poor QoL. Table 2 listed the significant associations with their QoL. Poor QoL among respondents was significantly associated with Malay ethnicity (p=<0.001), advancement trimester (p=0.002) and all the types of UI with stress UI, mixed UI and urge UI (p<0.001). (Page 10 Para 1)

Page 9 Line 4-6: Author may consider indicating if associations are associated with decreased or improved quality of life?

Authors’ comment: We thank for highlighting the grammatical issue, and we rephrase as below for better clarity.

“Pregnant women with mid to late trimester have three times odd with decreased QoL (OR 3.06, 95% CI 1.48-6.32). However, being Non-Malay, has improved QoL (OR 0.29, 95% CI 0.16-0.52). Unfortunately, for pregnant women who have stress UI, they have six times increased odds (OR 6.94, 95%CI 4.00-12.04) and those with urge UI have three times increased odds (OR3.87, 95%CI 0.48-31.28) to have a decreased QoL.” (Page 12 Para 2)

Page 13 line 13: This study……. with moderate UI …. negative Qol. This sentence needs to be reviewed / rephrased to improve clarity to the reader

Authors’ comment: We acknowledge for highlighting the grammatical issue, and we rephrase as below for better clarity.

“Sleep limitation were significantly associated among mid-late trimester respondents (p=0.044) and respondents from the high educational status (p=0.026). In contrast, being non-Malay, they were not associated with role limitation (p=0.007), physical limitation (p=0.012), social limitation (p=0.019) and severity measures (p<0.001). 

Among the three types of UI, mixed UI did not show any significant relationship with limitation to their personal relationship and emotion. Unfortunately, respondents with stress UI and urge UI has significant association with all domains of decreased QoL.”

(Page 13 Para 2-3)

Reviewer #2: 

First of all, thank you very much for give me the best opportunity for taking the reviewer role to review this manuscript. This is an interesting paper. The authors present a study to assess the risk factors and impact of UI on the QoL among incontinent pregnant women. A cross-sectional study design was used.

Author’s comment: We are delighted to get this feedback from the reviewer.

I have read carefully and found that this study is very carefully created and developed. Although this study has scientific interest, some important aspect should be reviewed by the authors. I hope that my opinions will help shape your research article more precise and interesting. The followings are my comments.

Author’s comment: We thank the reviewer’s expert opinion and guidance to shape this manuscript to improve the readers understanding and better citation.

1. The title of the study is not completely clear. The title indicates the dependent variable and samples. But, the independent variable and study design have not been included into the title. This makes the article sounded like a “review article” more than “research article”. Therefore, the authors should add the independent variable and study design in the title and rewrite the title.

Authors’ response: We thank the reviewer for the positive and encouraging feedback. We agreed with the suggestion. We would like to change the title to: 

“Quality of life among pregnant women with urinary incontinence: a cross-sectional study in a Malaysian primary care clinic.”

2. The authors presented an appropriate and clear detail about the abstract section, but some point should add in topics as following: the authors please add sample size of the participants and duration of data collection.

Authors’ response: We have added information regards to the sample size of this study. We wrote in the Results sections:

“Of the approached 610 respondents, 440 respondents consented to take part in the study, resulting in a response rate of 72.1%.” (Page 9 Para 1)

3. The authors used appropriate key words. However, the authors please add “risk factors” as the key words of the study to guide the reader to easily find a good research title and attract to read it.

Authors’ response: We thank the reviewer for this suggestion. We already added the keyword accordingly.

4. The introduction of the study is good written and shows the significance of the study. The author wrote the introduction in orderly manner beginning from relationship between pregnancy and urinary incontinence, definition, prevalence, risk factors, affect QoL, limitation of previous studies and aims of the study. Moreover, the authors presented a clear state of the aims of the study, but not showed the congruence with the aims of the study in the abstract section.

Authors’ comment: We thank and agree with your concern. We have corrected the introduction of the abstract accordingly.

“This study aimed to determine (1) the UI severity among pregnant women, (2) the QoL among pregnant women, and (3) the associations between UI and QoL among pregnant women attended primary care clinic. This study is reporting the third research objective from the previous study (10), and for the need assessment of our future mHealth app interventional study designed to improve UI among pregnant women (11).” (Page 5 Para 4)

5. The authors used a cross-sectional study as a study design. It is an appropriate design in this study. The setting is clear. The authors presented a clear state of sample, inclusion and exclusion criteria, duration of data collection, sample size calculation and sampling method. Moreover, the protection of human subjects was clearly stated and presented the number of IRB (JKEUPM-2019-297) in the section. All subjects gave written informed consent before entering the study.

Authors’ comment: We appreciate the supportive statements by the reviewer. Yes, we agreed that this study had been approved via the ethical committee before the study and all respondents consented to join this study.

6. The authors presented a clear state of the measurements of the study which included the details, scoring, and classification of all measurements in the study (ICIQ-UI SF and ICIQ-LUTSQoL). However, what is the reliability of ICIQ-UI SF and ICIQ-LUTSQoL in the study? The authors please state the reliability of all measurements in the study.

Authors’ comment: We apologize for not adding the information regarding the questionnaire's validity and reliability. We added under the Study instruments section as follow:

“A small pilot study was done among thirty pregnant women before this study to assess the validity and reliability of the Malay version questionnaire (13). The Cronbach alpha coefficient was 0.622 and 0.916 for the ICIQ-UI SF and ICIQ-LUTSqol respectively, which indicating a high level of reliability of this questionnaire in determining the quality of life.” (Page 8 Para 2)

7. In the results section, the authors presented all data both text and tables. At the beginning, the authors detailed about the demographic of the pregnant women with and without UI during pregnancy which presented in Table 1. Consequently, the authors used three tables to present the results of the study as following: Table 2 to present the scores of QoL, Table 3 to present seven domains of QoL with significant sociodemographics and Types of UI, and Table 4 to present the significant predictors of QoL among pregnant women. Therefore, the authors used appropriate number and running head title of the tables to report the significant findings, these may help the readers more clearly understand.

Authors’ comment: We appreciate your further elaborate on the presentation of our study results. We acknowledge with the reviewer’s comment that the Tables presentation can assist the reviewer understanding. Similarly, we hope the readers will be able to appreciate and comprehend the results of this study.

8. The discussion had been good written which show the consistency of the aims and the results of the study. Moreover, the authors discussed the results of the study comparing with the previous published studies. It may help the readers more clearly understanding.

Authors’ comment: We value the reviewer’s comment on the discussion section.

9. The authors clearly described the implication for practice of the study.

Author’s comment: We thank the reviewer’s statement regarding the study impact on the clinical practice. 

10. The authors did not state the strengths and limitations of the study. Please state the strengths and limitations of the study.

Author’s comment: We would like to thank you for this comment. We admit that we do miss to add the study limitation at the end of the discussion. We added the study limitation as below. 

“This study has been done in a single-centre primary care which affecting its external validity. The cross-sectional design declines the ability to conclude its temporal relationship. However, the study findings will be acceptable for our need assessment of future interventional study (12).” (Page 18 Para 4)

11. The authors clearly presented conclusion and followed logically from the results of the study.

Author’s comment: We would like to thank you for the reviewer’s statement and affirmation of this study's conclusion. 

12. The references that the authors cited in the text were published in the high standard journals in urogynecology field and had high relevance to the study which the authors interested in UI in the pregnant women. Moreover, most of references which the authors cited in the study were not over ten years. There were assumed that the knowledge from the previous published articles is not out of date. However, the authors please check the correct format of citations and references based on PLOS ONE.

Author’s comment: We would like to thank for reviewer’s statement in the references section. We tried to cite the recent and relevant articles in this manuscript. We have amend the style of the references according to the PLoSOne guideline.

Reviewer #3: 

I have used the STROBE statement to guide my review (Vandenbroucke JP, Von Elm E, Altman DG, Gøtzsche PC, Mulrow CD, Pocock SJ, Poole C, Schlesselman JJ, Egger M, Strobe Initiative. Strengthening the Reporting of Observational Studies in Epidemiology (STROBE): explanation and elaboration. PLoS Med. 2007 Oct 16;4(10):e297.)

Authors’ comment: We appreciate the reviewer sharing this guideline with us. We will use this to guide our manuscript.

Item 1a. study design not in title. Make it clear that the study investigates incontinence-specific QoL

Authors’ response: We thank the reviewer for the positive and encouraging feedback. We agreed with the suggestion. We would like to change the title to: 

“Quality of life among pregnant women with urinary incontinence: a cross-sectional study in a Malaysian primary care clinic.”

Item 1b. revise when comments addressed.

Item 2. Does not make it clear that there is QoL (as an ‘overall’ concept) and incontinence-specific QoL. When you report QoL data in the introduction it is not clear which of these you refer to. 

Authors’ comment: We acknowledge the reviewer’s concern and we have delete the paragraph and replace with:

“Therefore, pregnant women experiencing varieties of loss in their QoL reported with minimum impact (8) to significant impact (9) and socio-economic cost (6,7).” (Page 5 Para 3)

Paragraph 5 of the introduction says there are limited reports of UI in pregnant women in Asia. This is insufficient. What is the body of evidence on this topic in a similar population? I note at least one publication in the reference list that is directly relevant. Why is a study needed? What can your study add?

Authors’ comment: We acknowledge the reviewer’s insightful concern of the justification and added value of this study. This study is conducted for the need assessment of our future mHealth app interventional study designed to improve UI among pregnant women. We have add sentences as below:

“This study is reporting the third research objective from the previous study (10), and for the need assessment of our future mHealth app interventional study designed to improve UI among pregnant women (11).” (Page 5 Para 4)

Item 3. Study objective not clearly phrased. Seems to imply only interested in QoL in incontinent women, but the study cohort is women with and without incontinence. This is also the place to make it clear that you are investigating incontinence specific QoL

Authors’ comment: We acknowledge and agree the reviewer’s concern. We corrected as below: 

“This study aimed to determine (1) the UI severity among pregnant women, (2) the QoL among pregnant women, and (3) the associations between UI and QoL among pregnant women attended primary care clinic. This study is reporting the third research objective from the previous study (10), and for the need assessment of our future mHealth app interventional study designed to improve UI among pregnant women (11).” (Page 5 Para 4)

Items 7 and 8 and 11 and 16b. Please explain every variable (e.g. what are the sociodemographic and obstetric variables) and in what form the data were collected (e.g. date of birth), transformed (e.g. age at questionnaire completion) and categorised for analysis (e.g. dichotomisation into under 35 years of age, and 35 and over). Please also justify all categorisations (e.g. why 35?). Were all data self-reported including data such as gravidity and body weight and height (both of which you must have needed to calculate BMI) or were some data taken from the clinic record? Remember to discuss the accuracy of self-reported data (e.g. for weight/height) as this might influence your findings. 

Authors’ comment: We acknowledge the reviewer’s concern and we respond accordingly in the data collection.

The self-administered sociodemographic data for example age, ethnicity, occupation, monthly household income, level of education, Body Mass Index (BMI), number of pregnancies, number of the child alive and number of delivery (1) normal delivery, (2) vacuum, (3) forcep (4) Lower Segment Caesarean Section. The BMI was available as the respondents refer to their own antenatal records themselves. Age has been dichotomised with 35 years old based on the reference use (4)(L. Barbosa et al., 2018). (Page 7 Para 2)

The response categories on the ICIQ-LUTSQoL are ordinal yet the analysis seems is completed as if the data are continuous. Please explain/justify.

Authors’ comment: We understand that the reviewer would like to know more on the QoL scoring calculation. We added in the text with “ The seven domains are role limitations, physical limitations, social limitations, personal relationship, emotions, sleep, and severity measures according to the specific items in each domain.” (Page 13 Para 1)

Herewith, we are sharing the scoring calculation which received from the team ICIQ:

ICIQ-LUTSqol domain scores 

1) Role limitations

Score =(((Scores to Q 3A + 4A) – 2)/6) x 100

2) Physical limitations

Score =(((Scores to Q 5A + 6A) – 2)/6) x 100

3) Social limitations

[If 11A >/= 1] Score =(((Score to Q 7A + 8A + 11A) – 3)/9) X 100 

[If 11A = 0] Score =(((Score to Q 7A + 8A) – 2)/6) x 100 

4) Personal relationships

[If 9A+10A>=2]Score =(((Scores to Q 9A+10A) – 2)/6) x 100

[If 9A+10A=1] Score =(((Scores to Q 9A+10A) – 1)/3) x 100

[If 9A+10A=0] Treat as missing value

5) Emotions

Score =(((Score to Q 12A + 13A + 14A) – 3)/9) X 100 

6) Sleep / energy

Score =(((Scores to Q 15A + 16A) – 2)/6) x 100

7) Severity measures

Score =(((Scores to Q 17A + 18A + 19A + 20A) – 4)/12) x 100

9A, 10A and 11A “Not applicable” responses should be coded 0 for calculation purposes rather than 8 for validation purposes.

Item 12c. Were there any missing data? If yes, explain how this was handled. If none, I think it is worth saying so.

Authors’ comment: We have add “All were included into the analysis as there was no missing data.” (Page 9 Para 1)

Item 13. What do you know about the women who did not take part (e.g. demographics? Reasons for non-participation?)

Authors’ comment: We have add the reason for non-participation with “140 pregnant women refused to join the study, and 30 pregnant women repeated respondents.” (Page 9 Para 1)

Item 14. Table 1 presents the participant description by continence status (none, mild, moderate). Thus, there is no reporting of the descriptive data before it was categorised in some way.

Authors’ comment: We agree with this statement. We will add the citation to this results as it has been previously published. We add “Two-fifth (40.9%, n=180) of them reported with UI, 52.8% (n=95) reported slight UI, 44.4% (n=80) moderate UI and 2.8% (n=5) severe UI which the detail has been reported in our recent publication (10).” (Page 9 Para 2)

Item 16. The presentation of results in the text (the tables are generally good) lacks clarity throughout. For instance, the authors say “There was a clear association between the status of UI according to its severity and age, gravida, trimester, and vaginal delivery”. This tells me nothing about the direction of association, and UI status (e.g. is it younger or older age that is associated with having which UI severity?) 

And, in table 1 explain how the delivery mode data add up to n=440. If some of the women in this study are in their first pregnancy (n=170?) then they have not had a delivery yet. 

Authors’ comment: We acknowledge with this insightful feedback and we add “The total of 170 respondents was at their first-time pregnancy, and two-third of respondent had vaginal delivery (71.1%, n=192) and about a third of them (28.9%, n=78) had previous lower segment caesarean section (LSCS) before.” (Page 9 Para 3)

In tables 2 and 3 the mean difference and 95% CI for the difference is missing – p values are not sufficient. Without a mean difference and confidence interval it is very difficult to interpret this relative to the scoring of the QoL instrument. Why does table 2 not have delivery mode? 

Why are there a number of variables missing (e.g. age, education, etc) from table 3? This starts to appear like selective reporting. Why is age missing from the regression model? – based on table 2 it looks like it should be included.

Authors’ comment: We really appreciate for this comment and add the missing variable into the analysis. We re-structured our analysis and the Table 2, Table 3 and Table 4.

Items 18 to 21. Please address the study objective – clearly summarise the findings. For instance, ‘Non-Malay pregnant women, women in their second or third trimester, and women with stress UI, had greater odds of poor incontinence specific quality of life. Variables not associated with worse incontinence-specific quality of life were…………………The most affected QoL domains were……” 

Authors comment: We greatly appreciate this suggestion. We have add the sentences as below:

“Non-Malay pregnant women, women in their second or third trimester, and women with stress UI or urge UI, had greater odds of poor incontinence specific quality of life. Variables not associated with worse incontinence-specific quality of life were overweight or obese, high education and mixed UI. The most affected QoL domains were severity measures.” (Page 15 Para 1)

There is no discussion of any study limitations – what were they and what influence might they have on your findings? 

Author’s comment: We would like to thank you for this comment. We admit that we do miss to add the study limitation at the end of the discussion. We added the study limitation as below. 

“This study has been done in a single-centre primary care which affecting its external validity. The cross-sectional design declines the ability to conclude its temporal relationship. However, the study findings will be acceptable for our need assessment of future interventional study (12).” (Page 18 Para 4)

Please put the findings in the context of the data from other study on UI prevalence in pregnant women from Malaysia – what does your study add or confirm, and are there any inconsistencies and how might they be explained? Do the associations you find fit the pattern of associations others have found (or not)? How generalizable are these data? Most of the discussion about needs to inform women about various things irrelevant as your study objective is not about educating women or about interventions for UI. 

Authors’ comment: We appreciate this important view, and we have revise the discussion and conclusion. 

The implications for practice are not relevant to this study either. Only the first paragraph in the implications section could be considered appropriate – e.g. if it is common and affecting QoL then screening might be important – based on your data what can you say about who and how and when to find out if women have UI and what impact this is having for them?

Authors’ comment: We are grateful for the insightful suggestion. We add as below under “Implication for practice”:

“This study informs both the healthcare providers and pregnant women especially Malay ethnicity has poor QoL. Pregnant women with the advancement of trimester and from the educated background has sleep limitation. All pregnant women should be aware that they will experiencing sleep disturbance and to prepare measures to improve their sleep quality. Educated pregnant women who drives to work should be aware on their limitation and to ensure to have good quality of sleep to avoid road traffic accident when driving to work. 

Pregnant women with UI have experience limitation to all domain in QoL. The social limitation will add their stress level and risk of psychological distress. Their role limitation which may affect their ability to do their daily task and personal relationship limitation which may affecting their spouse. They need to spend more for the absorbent pad, and they worried of urine smell.

Therefore, both pregnant women and healthcare providers must consider to educate all pregnant women regarding importance of QoL and risk of UI during their pregnancy. UI as an essential condition to be dealt with clinically, instead of accepting it as a norm and addressing it by traditional means. Healthcare providers should screen the pregnant women even at their first trimester and among primigravida. All pregnant women should be aware on the UI and the conservative management available which is pelvic floor muscle training to enhance pregnant women's quality of life.” (Page 18, Para 1-3)

---

## [Decision Letter · Decision Letter 1]

13 Apr 2021

Quality of life among pregnant women with urinary incontinence: a cross-sectional study in a Malaysian primary care clinic.

PONE-D-20-35389R1

Dear Dr. Mohd-Sidik,

We’re pleased to inform you that your manuscript has been judged scientifically suitable for publication and will be formally accepted for publication once it meets all outstanding technical requirements.

Kind regards,

Peter F.W.M. Rosier, M.D. PhD

Academic Editor

PLOS ONE

Additional Editor Comments (optional):

Reviewers' comments:

Reviewer's Responses to Questions

**Comments to the Author**

1. If the authors have adequately addressed your comments raised in a previous round of review and you feel that this manuscript is now acceptable for publication, you may indicate that here to bypass the “Comments to the Author” section, enter your conflict of interest statement in the “Confidential to Editor” section, and submit your "Accept" recommendation.

Reviewer #1: All comments have been addressed

Reviewer #3: All comments have been addressed

2. Is the manuscript technically sound, and do the data support the conclusions?

Reviewer #1: Yes

Reviewer #3: Yes

3. Has the statistical analysis been performed appropriately and rigorously? 

Reviewer #1: I Don't Know

Reviewer #3: Yes

4. Have the authors made all data underlying the findings in their manuscript fully available?

Reviewer #1: Yes

Reviewer #3: Yes

5. Is the manuscript presented in an intelligible fashion and written in standard English?

Reviewer #1: Yes

Reviewer #3: No

6. Review Comments to the Author

Reviewer #1: The pilot study number of patients included for the validation of the questionnaire may not be adequate. However I feel if the statistician for Plos one can advise it may be still acceptable for the purpose of this publication.

Otherwise I am satisfied with the corrections

Reviewer #3: NONE FOR THE AUTHORS

WHAT FOLLOWS IS A COMMENT FOR THE EDITORS. While I have said 'accept' and all comments were addressed I feel the manuscript does need an edit for language/grammar to meet the editorial standards of PLOS One. While the meaning of the revisions can be discerned because I know what reviewer questions/concerns they are responding to, it would be much more difficult for a PLOS One reader to grasp the meaning. None of the editing would require a subject expert.

7. PLOS authors have the option to publish the peer review history of their article (what does this mean?). If published, this will include your full peer review and any attached files.

Reviewer #1: No

Reviewer #3: No

---

## [Editor Report · Acceptance letter]

19 Apr 2021

PONE-D-20-35389R1 

Quality of life among pregnant women with urinary incontinence: a cross-sectional study in a Malaysian primary care clinic 

Dear Dr. Mohd-Sidik:

I'm pleased to inform you that your manuscript has been deemed suitable for publication in PLOS ONE. Congratulations! Your manuscript is now with our production department. 

Kind regards, 

on behalf of

Dr. Peter F.W.M. Rosier 

Academic Editor

PLOS ONE